# Tailoring Endometrial Cancer Treatment Based on Molecular Pathology: Current Status and Possible Impacts on Systemic and Local Treatment

**DOI:** 10.3390/ijms25147742

**Published:** 2024-07-15

**Authors:** Pedro Ribeiro-Santos, Carolina Martins Vieira, Gilson Gabriel Viana Veloso, Giovanna Vieira Giannecchini, Martina Parenza Arenhardt, Larissa Müller Gomes, Pedro Zanuncio, Flávio Silva Brandão, Angélica Nogueira-Rodrigues

**Affiliations:** 1Oncoclínicas&Co—Medica Scientia Innovation Research (MEDSIR), São Paulo 04542-390, Brazil; 2Brazilian Group of Gynecologic Oncology (EVA), Rio de Janeiro 35500-025, Brazil; 3Department of Oncology, Santa Casa de Belo Horizonte, Belo Horizonte 30150-221, Brazil; 4Department of Radiotherapy, Hospital Beneficência Portuguesa de São Paulo, São Paulo 01323-001, Brazil; 5Department of Medicine, Federal University of Minas Gerais—UFMG, Belo Horizonte 30130-100, Brazil; 6DOM Oncologia, Belo Horizonte 30190-111, Brazil

**Keywords:** endometrial cancer, molecular pathology, local treatment, systemic therapies, precision medicine

## Abstract

Endometrial cancer (EC) is a heterogeneous disease with a rising incidence worldwide. The understanding of its molecular pathways has evolved substantially since The Cancer Genome Atlas (TCGA) stratified endometrial cancer into four subgroups regarding molecular features: *POLE* ultra-mutated, microsatellite instability (MSI) hypermutated, copy-number high with *TP53* mutations, and copy-number low with microsatellite stability, also known as nonspecific molecular subtype (NSMP). More recently, the International Federation of Gynecology and Obstetrics (FIGO) updated their staging classification to include information about *POLE* mutation and p53 status, as the prognosis differs according to these characteristics. Other biomarkers are being identified and their prognostic and predictive role in response to therapies are being evaluated. However, the incorporation of molecular aspects into treatment decision-making is challenging. This review explores the available data and future directions on tailoring treatment based on molecular subtypes, alongside the challenges associated with their testing.

## 1. Introduction

Endometrial cancer is a heterogeneous disease with an incidence of over 420,000 new cases diagnosed every year worldwide. It is the most common gynecological cancer in high-income countries and the sixth most common among females. EC incidence has been on the rise in high-income countries due to population aging and the increase in obesity, and it is no different in low- and middle-income countries [1]. Risk factors for EC include obesity, nulliparity, early age at menarche, use of hormone replacement, tamoxifen, and genetic variants (e.g., Lynch syndrome) [2,3].

The 5-year survival rate varies from 95% in localized disease to 18% in patients with distant metastasis [4]. Additionally, it ranks as the 19th most common cause of cancer-related death, with nearly 100,000 deaths yearly [1].

In 1983, Bokhman et al. classified EC into two subtypes: type 1, associated with obesity and hyperestrogenism and with a more favorable prognosis, and type 2, with unknown risk factors at that time and a worse prognosis [5]. Since then, the classification of EC has evolved substantially. Classification based on histology: endometrioid (around 80%) and non-endometrioid, including serous, clear cell, and carcinosarcoma [6], and other pathological findings such as grade, LSVI, depth of myometrial invasion, and presence of cervical invasion became essential to define the risk of recurrence and indicate the best adjuvant treatment following surgery [7].

However, over the past decade, new molecular research has become available and changed endometrial malignancy classification, showing that Bokhman’s dualism, though applicable, is oversimplified in explaining the complexity of the disease, as well as prognosticating and supporting treatment definition [8].

A better understanding of the molecular setting in EC became available over a decade ago, in 2013, with the TCGA publication [9], involving a stratification of EC as four distinguished subgroups [10], that yielded excellent prognostic assessments [11]: (A) *POLE* ultra-mutated, (B) microsatellite instability hypermutated, (C) copy-number high, with *TP53* mutations, and (D) copy-number low with microsatellite stability and intermediate prognosis [4,10,12]. Despite these identifications, it took some time for the leading societies and guidelines to use those subtypes in staging and adjuvant treatment indications, and the best way to do it still needs to be clarified [7,13]. In the advanced and metastatic setting, precision medicine is revolutionizing the treatment, targeting different mechanistic pathways based on molecular profiling [14]. Furthermore, beyond the TCGA’s four subtypes, more molecular granularity has been demonstrated, which has the potential to impact current treatment [15,16,17,18].

This review discusses the main impact of molecular pathology analyses on the clinical decision-making process, addressing current applications, future directions, and pitfalls in patients with endometrial cancer.

## 2. Molecular Assessment—From Bokhman’s to TCGA and Beyond

Endometrial carcinoma comprises a series of histological subtypes, each presenting different biological behaviors and responses to treatment. The endometrioid subtype is the most common, accounting for 75–80% of cases, while the other subtypes occur in smaller proportions. Carcinosarcoma is also classified as a subtype of carcinoma with a sarcomatoid component [19].

Exploring further into Bokmans’ classification, EC was classified based on biological behavior. Type I corresponded to most cases, with a more indolent behavior than type II. In this group, high estrogen exposure was a significant risk factor, whether the exposure was endogenous or exogenous. The primary representative of this group was grade 1–2 endometrioid carcinoma, which occurred mainly in perimenopausal or postmenopausal women. Additionally, the expression of estrogen and progesterone receptors was expected in this subset of tumors, which can be demonstrated in immunohistochemical examination and may have therapeutic targets, especially EC with more indolent behavior [5]. As for type II, the characterization consisted mainly of undifferentiated tumors, which were more aggressive and had a worse prognosis. They corresponded to 10–30% of endometrial carcinomas and usually occurred in older women. This group included serous, clear cell, and carcinosarcoma subtypes. However, some high-grade endometrioid carcinomas also fell into type II [20].

Molecular studies published after Bokhman’s dual classification have incorporated novel data on EC [21]. For instance, type I tumors are usually associated with PTEN, KRAS, CTNNB1, and PIK3CA mutations and microsatellite instability, as type II tumors are typically associated with human epidermal growth factor receptor (HER2) amplification and TP53 mutations [22]. PTEN alterations are an early event in the development of these neoplasms. The PI3K/PTEN/AKT/mTOR pathway and the RASERK pathway regulate the creation, cell growth, development, and apoptosis processes, with significant interaction between these pathways [23]. Germline mutations of *PTEN* lead to Cowden syndrome, with an increased risk of endometrial, breast, and thyroid neoplasms [24].

The TCGA publication in 2013 decoded endometrial cancer into four expanded subtypes as they have their characteristics, molecular features, and prognosis [25] (Table 1): (1) *POLE* ultra-mutated is characterized by a strong association with mutations in the exonuclease domain of polymerase-ε (*POLE*), a group of tumors with a good prognosis that could exclude the necessity of adjuvant treatment. (2) Microsatellite instability (MSI) hypermutated is characterized by a loss of DNA mismatch repair proteins (MLH1, MSH2, MSH6, and PMS2). (3) Copy-number high, with *TP53* mutations, is usually associated with unfavorable results/poor prognosis, and (4) copy-number low with microsatellite stability and intermediate prognosis [4,12,22].

Overall, approximately 8–10% of all endometrial cancers have *POLE* mutation, especially in young women with high-grade and early disease [25]. In the TCGA analysis, the ultra-mutated *POLE* group comprised 6.4% low-grade endometrioid carcinomas and 17.4% high-grade endometrioid carcinomas [22]. This subtype is characterized by high mutational rates (above 100 mutations per megabase) attributed to POLE mutations, and it has a high rate of 5-year recurrence-free survival rate for patients with grade 3 early endometrial carcinomas, above 96% [26].

Microsatellite instability occurs in 34–40% of cases of endometrial carcinoma and has an intermediate prognosis [25]. Mutations in the MLH1, MSH2, MSH6, and PMS2 genes are among the causes of MSI, and the observed tumor mutational burden is usually 10 to 100 mutations per megabase. Among the genes significantly mutated in this group are those known to be associated with endometrioid endometrial carcinomas, such as *PTEN*, *PI3K*, and *KRAS*. Up to 95% of cases present genomic alterations in the *PI3K–PIK3R1–PTEN* path. Lynch syndrome is characterized by germline mutations of genes that encode repair enzymes and occurs in 2–5% of EC [27]. It is recommended that the loss of expression of repair enzymes in all patients with endometrial carcinoma be investigated, especially among those with advanced disease, since there are relevant gains from the addition of immune checkpoint inhibitors (ICI) to first-line chemotherapy (CT) [16,18].

The *TP53* gene is an important tumor suppressor gene, and the p53 protein participates in cell cycle-checking processes. Genomic errors accumulate when its function is compromised and damage the apoptosis mechanism. Germline mutations of *TP53* also imply Li–Fraumeni syndrome [28]. *TP53*-mutated tumors may overexpress the human epidermal growth factor receptor (HER2) [25], while HER2 amplification occurs in up to 30% of cases [29]. In non-endometrioid carcinomas, mutations in *TP53* are common, as well as inactivation of p16 and changes in cell adhesion proteins, such as E-cadherin. In endometrial serous carcinomas, *TP53* mutation appears to be an early event in the tumorigenesis process and occurs in up to 90% of cases [29]. The *TP53* mutation can also be detected in other histological subtypes, including high-grade endometrioid carcinomas, associated with aggressive behavior and a more unfavorable prognosis [29].

Copy-number high tumors have many somatic alterations and low mutation rates, frequently associated with *TP53* mutations (9 in every 10 cases). Such tumors are considered as high-grade with poor prognosis [25]. The group with a high copy number, or serous-like, has the endometrial serous carcinoma as its primary representative. Other histology types also identified in this group are mixed carcinomas, high-grade endometrioid carcinomas, and even a tiny proportion of low-grade endometrioid carcinomas [30]. Up to a fifth of high-grade endometrioid carcinomas and up to 5% of low-grade endometrioid carcinomas have molecular characteristics like those of serous carcinomas. Moreover, it was confirmed that the profile of mutated genes that likely drive mutations varies depending on each subgroup [31].

Copy-number low tumors are considered to be cancers with no specific molecular profile (NSMP), low mutation burden, and low somatic copy-number alterations and prognosis related to the clinical stage in which the disease is diagnosed [25]. In this group with microsatellite stability, there are also wild-type *TP53* and non-mutated *POLE*. Here, mainly low-grade endometrioid carcinomas are identified, with 60% of these tumors being in this group. However, this also includes 25% of mixed carcinomas, 8.7% of high-grade endometrioid carcinomas, and 2.3% of serous carcinomas. In this group, changes in the PI3K pathway are also common, occurring in 92% of cases. Because of the heterogeneity of this group, future subsets could state a prognostic refinement, like the presence of mutations in exon 3 of β-catenin (*CTNNB1*) [25].

**Table 1 ijms-25-07742-t001:** Molecular classification of endometrial cancer. Adapted from [32,33].

Type	POLE (Ultramutated)	MSI (Hypermutated)/MMRd	Copy-Number Low (Endometrioid)/p53wt	Copy-Number High (Serous-like)/p53abn
Prevalence	7%	28%	39%	26%
Mutation frequency	Very high (>100 mutations/Mb)	High (10–100 mutations/Mb)	Low (<10 mutations/Mb)	Low (<10 mutations/Mb)
Commonly mutated genes	*POLE* (100%) *DMD* (100%) *CSMD1* (100%) *FAT4* (100%) *PTEN* (94%)	*PTEN* (88%) *PIK3CA* (54%) *PIK3R1* (42%) *RPL22* (37%) *ARID1A* (37%)	*PTEN* (77%) *PIK3CA* (53%) *CTNNB1* (52%) *ARID1A* (42%) *PIK3R1* (33%)	*TP53* (92%) *PIK3CA* (47%) *FBXW7* (22%) *PPP2R1A* (22%) *PTEN* (10%)
Copy number aberrations	Very low	Low	Low	High
MSI/MLH1 methylation	Mixed High and Low MSI, stable	High MSI (MLH1, PMS2, MSH2, and/or MSH6 deficiency)	MSI stable	MSI stable
Histological subtype	Endometrioid	Mostly endometrioid	Endometrioid	Serous, 25% high-grade endometrioid and mixed G3
Grade	G1-G3	G1-G3	G1-G2	G3
Other features	Ambiguous histomorphology Dense immune infiltrates	Display tumor infiltrate lymphocytes	CTNNB mutations are associated with poor prognosis Subgroup with amplification of chromosome arm 1p has poor prognosis	Similar to high-grade serous ovarian cancer L1CAM expression associated with poor prognosis
Prognosis	Good	Moderate	Moderate	Poor
Diagnostic test	Sanger/NGS Tumor mutation burden	MMR-IHC (MLH1, MSH2, MSH6, PMS2) MSI assay Tumor mutation burden		p53-IHC NGS Somatic copy-number aberrations

Microsatellite instability (MSI), L1 cell adhesion molecule (L1CAM), immunohistochemistry (IHC), next-generation sequencing (NGS).

It is worth mentioning that nearly 3–6% of all endometrial cancers are classified with more than one of these molecular subgroups. Additional molecular risk factors have been studied lately, aiming to define new prognostic factors, such as L1-cell adhesion molecule (L1CAM) overexpression, which is strongly associated with p53 mutations and is an independent risk factor for loco-regional and distant disease [25].

Also, it is essential to address that although molecular testing has become fundamental for clinical decision-making in EC, cost-effectiveness remains an important pitfall for accessibility. Previous data have already reported that tumor molecular testing (TMT) could be cost-saving with equivalent effectiveness against no testing at all at a willingness-to-pay threshold of USD 100,000/quality-adjusted life-year (QALY) gained [30]. However, when TMT is compared to mismatch repair IHC costs alone, TMT costs USD 182,798/QALY more, becoming economically unfavorable [30].

In this perspective, Talhouk et al. [34] developed a more pragmatic testing method for molecular assessment of EC, known as Proactive Molecular Risk Classifier for Endometrial Cancer (ProMisE), intending to create a clinically applicable molecular-based classification system for EC [35], based on MMR protein search (MLH1, MSH2, MSH6, PMS2) by immunohistochemistry, POLE mutational analysis, and p53 immunohistochemistry as a surrogate for ‘copy-number’ status instead of genomic methodology [34]. Later, a second cohort validated the results found in the previous paper [36]. Final ProMisE (Figure 1) results showed that POLE had a better prognosis, and p53 mutants had the worst [11].

In the Leiden/TransPORTEC study, immunohistochemistry, sequencing for p53, immunohistochemistry to evaluate microsatellite instability, and sequencing for POLE were performed. The tests’ prognostic role was also demonstrated in a cohort of 947 patients [37,38].

There are also uncertainties about the performance of IHC and MSI in the analysis of endometrial tumors since there is a need for clinical validity and cost-effectiveness criteria before performing these tests [39]. Testing for mismatch repair (MMR) by IHC is much cheaper and more accessible than MSI testing by PCR, with similar accuracy. Moreover, performing both tests simultaneously is futile due to the high concordance among these two tests. PCR should be performed solely when MMR testing is unclear. Furthermore, it is suitable for reducing costs with MMR IHC since a two-step test has similar accuracy to a single-step four-antibody test. In that matter, it is suggested that an mismatch repair (MMR) initial panel with *PMS2* and *MSH6* must be tested, and, in cases of any defects detected, a second test that includes *MLH1* and *MSH2* should be performed [40].

The information brought by the molecular biology of EC may be relevant as a tool for predictive assessment and in the definition of therapeutic strategies, and the new FIGO 2023 document supports its incorporation in its staging system, despite controversies [13]. According to the new FIGO staging, in cases of anatomical stage I–II, the detection of a *POLE* mutation currently results in a decrease in the FIGO stage [13]. At the same time, abnormal p53 leads to increased staging, yielding escalation or de-escalation of adjuvant treatment. Furthermore, in cases of advanced disease, molecular types and their associations with hormonal expression, loss of expression of repair enzymes, and overexpression of HER2 are also valuable for choosing therapies.

## 3. Biomarkers and Therapeutic Targets

In recent years, biomarkers have become essential for guiding treatment decisions for patients with endometrial cancer [41,42].

The presence of estrogen receptor (ER) and progesterone receptor (PR) in tumor tissue is routinely evaluated with immunohistochemistry (IHC). However, there is no standard cut-off value for positivity, and receptor positivity in EC is commonly adopted from breast cancer studies [25]. Usually, early-stage, well-differentiated endometrial cancers are positive for ER and PR, while advanced-stage, poorly differentiated tumors often lack one or both hormone receptors [43].

Hormonal therapy has been used in the treatment of uterine-limited disease not suitable for primary surgery or for patients desiring uterine preservation for fertility and in the recurrent and metastatic setting in those tumors that are indolent, low-grade, or in patients for which other therapeutic modalities may be too toxic since it is well tolerated and has response rates ranging from 9 to 33% [44]. Progestins, such as medroxyprogesterone and megestrol acetate, are the recommended agents, but other options include aromatase inhibitors, fulvestrant, and combined progestin agents with tamoxifen [44,45]. Responses have been reported in ER/PR-negative tumors, but higher responses are seen in hormone-positive tumors [7].

Emerging data from phase II trials support a combination of hormonal therapy with cyclin-dependent kinases 4 and 6 (CDK4/6) inhibitors in patients with metastatic or recurrent ER-positive EC. In the NSGO-PALEO/ENGOT-EN3 trial, 73 patients with ER-positive endometrioid advanced EC were randomized for letrozole with either palbociclib or placebo, with significant improvement in progression free survival (PFS)—median 8.3 versus 3.0 months, respectively, (hazard ratio [HR] 0.56–95% CI 0.32 to 0.98; *p* = 0.041) [46]. In another trial, 30 patients with recurrent ER-positive EC (28 with endometrioid histology) received letrozole in association with abemaciclib, resulting in a median PFS of 9.1 months, an objective response rate (ORR) of 30%, and a median duration of response of 7.4 months [47].

Mutations in the phosphatase and tensin homologue deleted on chromosome 10 (PTEN)/phosphatidyl- inositol 3-kinase (PI3K)/protein kinase B (Akt)/mammalian target of rapamycin (mTOR) pathway is frequently seen in EC, so to increase response rates and mitigate resistance, hormonal therapy has also been studied in association with everolimus, an mTOR inhibitor [48]. In a phase II trial, everolimus, in combination with letrozole, improved median PFS when compared to medroxyprogesterone acetate or tamoxifen—six months and four months, respectively. This benefit was more pronounced in chemotherapy-naïve patients, with a 28-month median PFS [49].

Evaluating the mismatch repair proteins is essential in EC. Around 25–30% of patients have mismatch repair deficiency (MMRd)/microsatellite instability-high (MSI-H), and recurrent or metastatic endometrioid ECs exhibit around 7% higher frequencies of MSI-H/MMR-D compared to matched primary tumors [50,51]. MMR status has prognostic implications and can be used as a marker for selecting therapy with immune checkpoint blockade in the setting of advanced disease [7]. The use of immunotherapy in cancer will be discussed further.

Another biomarker is the human epidermal growth factor receptor 2 (HER2). The HER2 gene is amplified in 17–33% of carcinosarcoma, uterine serous carcinoma, and a subset of high-grade endometrioid endometrial cancer [52]. In a randomized phase II trial, patients with stage III or IV or recurrent HER2-positive uterine serous carcinoma were randomly assigned to receive carboplatin-paclitaxel for six cycles with or without intravenous trastuzumab until progression or unacceptable toxicity, with improvement in median PFS of 8.0 months in the control arm versus 12.6 months in the group that received trastuzumab (HR 0.44; 90% CI 0.26 to 0.76; *p* = 0.005) [53].

HER2-expressing EC was also evaluated in the phase II study DESTINY-PanTumor02. In this trial, patients with locally advanced or metastatic solid tumors that overexpress HER2 (IHQ 3+ or 2+) that had worsened after at least one systemic treatment or had no treatment options were treated with trastuzumab deruxtecan (T-DXd). In the cohort, the ORR was 57.5% for all patients, 84.6% for those with IHC 3+, and 47.1% for IHC 2+ [54]. With the results of these studies, as of April 2024, the Food and Drug Administration (FDA) granted accelerated approval of T-DXd for patients with unresectable or metastatic solid tumors HER2 3+ by IHC that received prior systemic therapies and have no alternative treatment options [55].

ENGOT-EN5/GOG-3055/SIENDO was a phase III trial in which 263 patients with advanced or recurrent EC that presented partial or complete response after one line of taxane-platinum CT were randomized for selinexor versus placebo as maintenance treatment [56]. Selinexor is a specific inhibitor of exportin 1 (XPO1), leading to nuclear activation of suppressor and regulatory proteins, such as p53 [57]. The study did not reach statistical significance in the intention-to-treat population, with a median PFS of 5.7 months for selinexor versus 3.8 months for the control arm (HR 0.70; *p* = 0.024). However, in a subgroup analysis according to molecular classification, selinexor showed substantial improvement in median PFS amongst patients with TP53 wildtype—13.7 months versus 3.7 months (HR 0.375; 95% CI 0.210–0.670; *p* = 0.0003), and with NSMP (median PFS NR and 3.71 months; HR 0.163; 95% CI 0.060–0.444; *p* < 0.0001). Another molecular category that demonstrated benefit with selinexor was the MSS/pMMR subgroup, with a median PFS of 6.9 months in the experimental arm against 5.4 months in the control arm (HR 0.593; 95% CI 0.388–0.905; *p* = 0.007). This suggests that the maintenance therapy may be promising and can be further explored in these molecular subtypes [56].

Despite limited data on its prevalence, homologous recombination deficiency (HRD) is also a potential biomarker in EC. Still, tumors with a deficiency in the homologous recombination pathway are likely to benefit from poly-ADP ribose polymerase (PARP) inhibitors [58]. The trials RUBY part 2 and DUO-E evaluated the association of immunotherapy to chemotherapy with the addition of PARP inhibitors during the maintenance treatment.

Part 2 of the RUBY trial assessed the efficacy and safety of adding niraparib to dostarlimab as a maintenance treatment for patients with recurrent or primary advanced stage III or IV EC. Dostarlimab plus CT followed by dostarlimab plus niraparib compared to placebo plus CT followed by placebo showed improvements in median PFS in the overall population (14.5 months vs. 8.3 months) and in the pMMR/MSS population (14.3 months vs. 8.3 months) [59]. 

DUO-E was a three-arm study: carboplatin/paclitaxel plus placebo followed by placebo maintenance; chemotherapy plus durvalumab with durvalumab maintenance until disease progression; and carboplatin/paclitaxel plus durvalumab followed by maintenance durvalumab plus olaparib. The durvalumab + olaparib arm had a statistically significant 45% lower risk of disease progression or death when compared to the control arm, and the addition of the PARP inhibitor demonstrated a more pronounced benefit in the pMMR population (HR for durvalumab + olaparib versus control 0.57; 95% CI 0.44 to 0.73) [18].

In the era of precision medicine, the management of EC should be based on individualized risk assessment molecular and biomarker subtyping to help guide treatment and improve patient outcomes.

## 4. Current and Future Impact on Surgery

Precision medicine through molecular profiling has taken a prominent role in treating solid tumors, and it is widely expected that this will continue to expand. Considering EC, the molecular classification system has now been incorporated into virtually every guideline available, and molecular-directed treatment strategies are currently being researched, presumably leading to a further transformation of its treatment paradigm [60].

EC molecular subtypes represent a functional classification system that evaluates tumor characteristics. They have the potential to predict the prognosis of patients who underwent initial surgery and determine the usefulness of appropriate molecular-targeted therapy for patients with recurrence or progression. Molecular classification can distinguish patients with similar histological features but different prognoses and guide therapeutic strategies and appropriate surveillance [61].

Now that guidelines have drastically been modified according to the molecular classification system, the time has come for the next phase: to determine its implications for surgical management and explore whether targeted therapy directed at these molecular subgroups will result in better disease outcomes [60].

The standard management of EC involves surgery, chemotherapy, and/or radiation therapy. The gold standard staging procedure for EC is total hysterectomy with bilateral salpingo-oophorectomy (TH/BSO) with, if necessary, lymph node surgical assessment [62]. In some selected premenopausal patients, ovary preservation may be a safe choice in stage I endometrioid cancer [63]. Minimally invasive surgery does not compromise oncological outcomes and has a lower rate of complications, so it should be proposed in patients with macroscopically uterine-confined cancer. LAP2 trial compared oncological outcomes in laparoscopic (LPS) vs. laparotomic (LPT) surgery, showing recurrence rates of 11.4% for LPS versus 10.2% for LPT surgery and a 5-year overall survival rate of up to 84.8% [64,65]. Robotic surgery may be the surgical choice for the severely obese and for patients at higher anesthesiologic risk [66]. During the surgery, suspicious intraperitoneal areas and enlarged lymph nodes should be biopsied, and peritoneal cytology should be collected. Through surgical staging, an accurate diagnosis, extension of the disease, and prognostic assessment can be defined, and patients who require further adjuvant therapy can be selected. Routine lymph node dissection identifies patients with nodal localization requiring adjuvant treatment with radio and/or chemotherapy [67,68,69]. Guidelines recommend sentinel lymph node biopsy in patients with low-risk and intermediate-risk diseases [70].

In the era of the new molecular classification of EC, questions have arisen about its implementation not only in the planning of the adjuvant treatment but also in the surgery planning and especially the lymph node staging [71]. In 2016, Talhouk et al. showed that molecular classification from the pre-surgery endometrial samples can accurately predict the molecular features of the final hysterectomy tumor, with even higher concordance than grade and histology [72]. During the initial diagnosis, this information could alter the surgical management plan and help choose patients who will undergo fertility-sparing [73] carefully. Patients with favorable molecular features could be spared from any lymph node staging technique, and high-risk patients could be offered more radical surgical lymph node staging [74].

Asami et al. demonstrated that among 265 patients who underwent initial surgery, classified according to immunohistochemistry, patients with DNA polymerase epsilon exonuclease domain mutation had an excellent prognosis, patients with no specific molecular profile and mismatch repair protein deficiency had an intermediate prognosis. Those with protein 53 abnormal expression (p53abn) had the worst prognosis (*p* < 0.001). In the NSMP group, mutant *KRAS* and wild-type *ARID1A* were associated with significantly poorer 5-year RFS (41.2%) than other genomic characteristics (*p* < 0.001). The distribution of the subtypes differed substantially between patients with recurrence/progression and classified by sequencing (*n* = 764) and patients who underwent initial surgery (*p* < 0.001). Among patients with recurrence/progression, 51.4% had the opportunity to receive molecular-targeted therapy [61].

Another study showed that ovarian infiltration by endometrial cancer was associated with molecular profile. Of 317 patients with EC who underwent bilateral oophorectomy, 27 (9%) had malignant ovarian tumors, of whom 11 (41%) had no gross ovarian involvement on an intraoperative survey. For patients with sequencing, concurrent malignant ovarian tumors were diagnosed in 0/14 (0%) *POLE*, 2/48 (4%) copy number-low/no specific molecular profile, 10/22 (45%) microsatellite instability-high, and 3/6 (50%) copy number-low/*TP53* abnormal patients (*p* < 0.001). The authors concluded that the integration of molecular and pathologic data may improve risk stratification of pre-menopausal patients with EC and enhance candidate selection for ovarian preservation [75].

The European Society of Gynecological Oncology (ESGO), the European Society for Radiotherapy and Oncology (ESTRO), and the European Society of Pathology (ESP) guideline has also highlighted the importance of work-up for fertility preservation treatments and the management and follow-up of fertility preservation in young patients. Fertility-sparing treatment (FST) in EC could be an option for a subgroup of women selected based on a thorough evaluation of their reproductive potential. In such cases, molecular classification could be helpful [76].

However, further trials are necessary to clarify the role of EC molecular classification in surgical approaches, such as the EUGENIE trial, which is currently recruiting [77]. This prospective trial examines to what extent molecular classification will guide surgical staging. The primary endpoint is set as the number and site of metastasis in each molecular subgroup, whereas secondary endpoints include time to recurrence and overall survival. Results on staging and oncological outcomes are expected in 2027 and 2029, respectively.

## 5. Current and Future Impact in the Adjuvant Systemic Therapy

Recommendations for adjuvant therapy for EC have been determined by each patient’s risk of disease recurrence, considering clinicopathological factors such as age, stage, histological subtype, tumor grade, and LSVI [78]. Some studies have demonstrated the improvement in risk assessment by integrating molecular and clinicopathological factors in early-stage endometrial carcinoma [7]. Despite the absence of randomized controlled trials that provide definitive guidance on the use of adjuvant therapy based on the molecular profile in EC, some retrospective analyses have consistently demonstrated the potential of this approach [7]. The ESGO, ESTRO and ESP published a guideline with an integrated classification system with molecular and clinicopathological features to guide adjuvant treatment choices in 2020 [79]. In 2023, FIGO updated the staging system including information about the *POLE* mutation and p53 status [13].

In the PORTEC-3 trial, molecular data were reported on 410 high-risk patients receiving adjuvant radiotherapy with and without chemotherapy. Patients with a p53abn EC had poor prognosis and a statistically significant benefit from combined adjuvant concurrent chemoradiotherapy (CTRT) with an absolute difference of 22.4% for relapse-free survival (RFS) and 23.1% for overall survival (OS). The outcomes for the POLEmut group were excellent, but all patients received adjuvant treatment, as the control group included external radiotherapy (RT). Only one patient with a POLEmut EC had disease recurrence, resulting in a 5-year RFS and OS of 100% with CTRT versus 96.6% with RT. No benefit was observed from CTRT versus RT alone in patients with MMRd EC. NSMP EC patients showed a tendency for CTRT benefit, similar to the overall trial results, but additional studies are required to determine the role of chemotherapy in this subgroup [80].

A retrospective analysis of 2427 endometrial cancers showed that in patients with MMRd EC, there was no benefit in disease-specific survival (DSS) or PFS with the addition of chemotherapy compared to radiation alone in European Society for Medical Oncology (ESMO) classification of high-risk (*p* = 0.694) or ESMO high, advanced, metastatic risk groups combined (*p* = 0.852). In patients with p53abn endometrial cancer, adjuvant CT given with radiation was associated with significantly longer DSS compared to radiation alone in ESMO high-risk (*p* = 0.007) and ESMO high, advanced, and metastatic risk groups combined (*p* = 0.015), even when restricted to the stage I disease (*p* < 0.001) and when compared in serous vs. non-serous histotypes (*p* = 0.009) [81]. 

The NRG/GOG0210 study’s exploratory analysis showed that adjuvant treatment did not significantly affect PFS in the MMRd patients’ group. There was a trend of improvement in PFS (HR 0.24—95% CI 0.05–1.16, *p* = 0.07) only for cases of probable MMRd (defined as positive MSI and/or IHC defect with the absence of MLH1 methylation) [82].

A retrospective study evaluated the prognostic role of molecular classification in patients with high-grade endometrial cancer who underwent lymphadenectomy and did not receive adjuvant treatment. Five-year recurrence rates were 36.7% for women with p53abn EC, 0.0% for POLEmut EC, 13.4% for MMRd EC and 42.9% for NSMP EC (*p* < 0.001). Patients with p53abn endometrial cancer had a poor clinical outcome, even if lymph node-negative and stage I. Among patients without adjuvant treatment (*n* = 264), none with POLEmut EC (*n* = 26) had a recurrence. Substantial LVSI was a significant prognostic factor for recurrence and OS, independent of the complete cohort’s molecular subgroups and other clinicopathological features [83].

A meta-analysis based on individual patient data assessing treatment effects in patients with POLEmut EC showed no benefit from adjuvant treatment. Most patients (87%) had a low or intermediate-risk endometrial cancer by ESMO 2013 criteria. Clinicopathological factors such as age, histology, grade, and LVI did not seem to have the same relevance in the POLEmut subtype; only the stage was associated with higher recurrence or death [84].

The *ESGO*/*ESTRO*/*ESP* guideline recommends considering the omission of adjuvant treatment for patients with endometrial carcinoma stage I, II, and pathogenic POLE mutation [79]. Despite these retrospective data, prospective evaluation of the molecular characteristics in randomized trials is highly recommended.

The Rainbo Research Consortium developed a program of four clinical trials to evaluate adjuvant treatment according to molecular classification. The p53abn-RED trial is a phase III trial in which patients with invasive stage I–III p53abn endometrial cancer are randomized to adjuvant chemoradiation followed by olaparib (300 mg twice daily, orally) for two years or adjuvant chemoradiation alone. The MMRd-GREEN trial is a phase III trial in which women with stage II LVSI or stage III MMRd endometrial cancer are randomized to adjuvant pelvic radiotherapy combined with and followed by the PD-L1 inhibitor durvalumab (13 cycles of 1500 mg intravenously, every 4 weeks) for one year or radiotherapy alone. The NSMP-ORANGE trial is a phase III non-inferiority trial in which patients with stage II with LVSI or stage III NSMP EC are randomized to adjuvant radiotherapy followed by progesterone tablets for two years or adjuvant chemoradiation alone. The POLEmut-BLUE trial is a phase II trial assessing the safety of de-escalation of adjuvant therapy: no adjuvant therapy for select stage I–II diseases and no adjuvant treatment or radiotherapy only for higher-risk stage I–III disease. All trials except red are recruiting; these results will help the decision-making process regarding adjuvant treatment [85].

## 6. Current and Future Impact on Radiotherapy

Radiotherapy is an essential adjuvant treatment in endometrial cancer, mainly in reducing the risk of locoregional recurrence. The most relevant studies with adjuvant radiotherapy are based only on clinical and pathologic risk factors, including age, pathologic grade, histologic subtype, depth of myometrial invasion, LVSI, tumor stage (T) and lymph node metastasis (N). In GOG—the 99 study, intermediate–high risk was defined if patients were 50 years of age or older and had at least two risk factors (grade 2 or 3 disease, LVSI, or invasion of the outer one-third of the myometrium) or 70 years of age or older and at least one risk factor. This trial showed a statistically significant reduction in the incidence of locoregional recurrence with adjuvant radiotherapy compared to observation in this population [86]. GOG-99, PORTEC-1, and PORTEC-2 have used the FIGO 1988 staging system to define their inclusion criteria [86,87,88,89,90]. More recent clinical trials, such as GOG-249, GOG-258, and PORTEC-3, were based on the FIGO 2009 staging system [91,92,93]. 

In 2023, FIGO published its new staging system incorporating molecular classifications and histology. This new staging includes *POLEmut* and p53abn in FIGO stage I and II classification. However, *POLEmut* and p53abn do not modify FIGO stage III and IV classification, while the status of MMRd and NSMP do not change FIGO staging [94]. The new FIGO staging system presents many differences from the previous ones (FIGO 1988 and FIGO 2009), making it challenging to extrapolate data from old studies in this new scenario.

Advances in molecular profile help us to better prognosticate patients with endometrial cancer and select adjuvant radiotherapy more appropriately, reducing undertreatment and overtreatment. In 2023, Horeweg et al. published data from intermediate-risk patients enrolled in PORTEC-1 and 2, finding the prognostic significance of *POLEmut* and p53abn [95]. Recently, León-Castillo et al. reported the impact on prognosis and benefit from adjuvant chemoradiotherapy in RFS compared to radiotherapy alone for patients with high-risk disease and p53abn. In contrast, there was no difference between the two arms in patients with *POLEmut* disease and excellent outcomes for both [80].

The ESGO/ESTRO/ESP and American Society for Radiation Oncology (ASTRO) published their respective guidelines and corroborated the use of novel molecular classification for all patients with endometrial carcinomas, including *POLEmut*, MMRd/NSMP, and p53abn, into the definition of prognostic risk groups [79,96].

Despite this, we must be careful to use these molecular factors to exclusively define treatment strategy in the current scenario, as we still need prospective and randomized phase III trials with solid results in this context.

## 7. Current and Future Impact on the Systemic Treatment of the Advanced Disease

Systemic treatment selection is guided by prior treatment, patient, and disease characteristics (de novo or recurrent), and molecular panel. The standard treatment for years has been based on the GOG 209 study, a phase III non-inferiority trial, comparing the combination of carboplatin and paclitaxel versus cisplatin, doxorubicin, and paclitaxel, demonstrating that the combination of carboplatin and paclitaxel was non-inferior to the three-drug regimen in terms of OS, PFS and less toxic [97].

Regarding the addition of immunotherapy in stages III and IV, so far, four phase III studies assessing the role of immunotherapy (pembrolizumab, dostarlimab, atezolizumab and durvalumab) concomitantly with CT (platinum doublet) followed by maintenance with immunotherapy in first line have been published. RUBY trial—part I evaluated the use of dostarlimab concomitantly with CT for six cycles, followed by maintenance of dostarlimab for three years. Eligible patients were diagnosed with stage III or IV or recurrent EC, and aggressive histology were also included, such as carcinosarcoma, serous, and clear cell histology. The use of immunotherapy presented a statistically significant reduction in the risk of death by 31% (hazard ratio [HR] = 0.69, 95% confidence interval [CI] = 0.539–0.890) and a significant improvement of 16.4 months in median overall survival (44.6 months vs. 28.2 months). In a prespecified exploratory analysis of the pMMR/MSS population, dostarlimab plus chemotherapy vs. chemotherapy alone showed a trend in reduced risk of death by 21% (HR = 0.79, 95% CI = 0.602–1.044) and a clinically meaningful improvement of 7 months in median overall survival (34.0 months vs. 27.0 months). Furthermore, in subgroup analysis, there was no benefit in adding dostarlimab to QT in stage III compared to the evident benefit of the combination in stages IV and recurrent [16].

An exploratory analysis from the RUBY trial by molecular classification was performed in 400 patients with whole-exome sequencing results available; PFS according to molecular subgroup were hazard ratio 0.31 (95% CI, 0.17 to 0.56), 0.55 (95% CI, 0.3 to 0.99), 0.77 (95% CI, 0.55 to 1.07) in the MMRd/MSI-H, p53 abnormal, and non-specific molecular profile subgroups, respectively [98].

The second phase III trial to evaluate immunotherapy combined with CT is NRG 018, which associated pembrolizumab with six cycles of carboplatin and paclitaxel followed by maintenance pembrolizumab for two years, including stage III or IVA with measurable disease, stage IVB with or without measurable disease, or relapsed disease. Two distinct cohorts of women were included: dMMR and pMMR patients. PFS at 12 months was 74% for dMMR in the pembrolizumab group versus 38% in the placebo group (HR = 0.30. 95% CI: 0.19–0.48—*p* < 0.001). In the pMMR population, PFS was 13.1 months with pembrolizumab versus 8.7 months with placebo (HR = 0.54. 95% CI: 0.41–0.71—*p* < 0.001) [99].

In the AtTEnd study, patients were randomized between atezolizumab, or placebo associated with CT followed by atezolizumab or maintenance placebo until disease progression. Median PFS in the dMMR population was 6.9 months in the placebo group versus not reached in the atezolizumab group (HR = 0.36. 95% CI: 0.23–0.57—*p* = 0.0005), with a tendency to gain OS (data were still immature in this interim analysis) [100].

In the DUO-E study, durvalumab was added to CT and used as maintenance until disease progression, alone or concomitantly with olaparib. The use of durvalumab with or without olaparib demonstrated a gain in PFS in the intention-to-treat population, with a median PFS of 9.6 months in the control group, 10.2 months in the durvalumab arm (HR versus control = 0.71. 95% CI: 0.57–0.89; *p* = 0.003) and 15.1 months in the durvalumab arm associated with olaparib (HR versus control = 0.55. 95% CI: 0.43–0.69—*p* < 0.0001). OS data are still immature but also suggest a gain in the experimental arms. Subgroup analysis indicates that the PFS benefit with durvalumab was more significant in the dMMR group. However, adding olaparib to durvalumab demonstrated a more significant PFS benefit in the pMMR population [18].

In some subgroups, such as serous endometrial cancers overexpressing HER2, the addition of trastuzumab to frontline CT is recommended as a maintenance until progression, based on a randomized phase II trial, patients stage III-IV or recurrent HER2-positive uterine papillary serous cancer were randomly assigned to carboplatin/paclitaxel for six cycles, with or without intravenous trastuzumab, until progression or unacceptable toxicity. Median PFS was 8 (control) versus 12.9 months (experimental; hazard ratio [HR] 0.46, 90% CI 0.28–0.76). OS was 24.4 months in the control group and 29.6 months in the trastuzumab group (HR 0.58, 90% CI 0.34–0.99). Recently, the Pan-destiny tumor-02 trial was presented for HER2-expressing (immunohistochemistry [IHC] 3+/2+ by local or central testing) locally advanced or metastatic disease after ≥1 systemic treatment or without alternative treatments, showing in the endometrial cancer cohort with the use of trastuzumab-deruxtecan 5.4 mg/kg q3 weeks, durable clinical benefit, meaningful survival outcomes, and safety [15].

## 8. Conclusions

The management of endometrial cancer is rapidly evolving, leveraged by the understanding of its molecular biology and the application of precision medicine. The most updated staging systems and guidelines are beginning to integrate molecular classification into (FIGO/ESMO) practice, and targeted therapies are being developed and approved at an increased rate.

Several challenges remain, such as the absence of biomarkers and specific targets in a subset of patients, especially in the NSMP subgroup, which presents heterogeneously and could be classified based on its molecular landscape with different clinical settings and prognostics.

Another challenge is how to sequence therapies in the advanced setting. No prospective data support the use of immunotherapy after the progression on immunotherapy, and the best moment to use antibody conjugates is still unclear.

Finally, there are considerable disparities in endometrial cancer management worldwide. Affordable and reproducible models of care are needed, especially in low- and middle-income countries, with access to precision medicine tests, new therapies, and clinical trials.

## Figures and Tables

**Figure 1 ijms-25-07742-f001:**
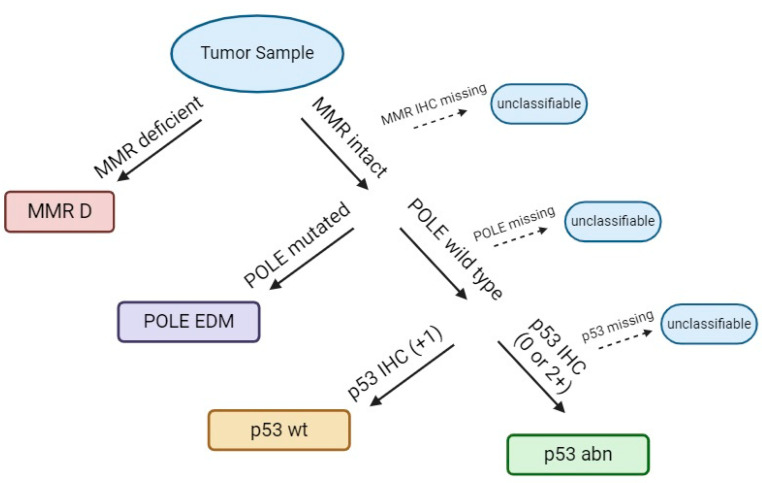
ProMisE algorithm. Adapted from the ProMisE algorithm [34].

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
