# Peer review of "Tailoring Endometrial Cancer Treatment Based on Molecular Pathology: Current Status and Possible Impacts on Systemic and Local Treatment"

_ijms, 2024, doi:10.3390/ijms25147742_

Round 1
Reviewer 1 Report
Comments and Suggestions for Authors
I have reviewed your manuscript Tailoring endometrial cancer treatment based on molecular pathology: current status and possible impacts on systemic and local treatment.
The manuscript aims to review the current status and future directions of tailoring endometrial cancer (EC) treatment based on molecular pathology, focusing on the implications for systemic and local treatments.
The topic is highly relevant , but less original as previous manuscripts and literature reviews on this topic have benn published.
This review provides a comprehensive summary of the current knowledge on EC molecular subtypes and their prognostic and predictive roles. It offers valuable insights into the potential for personalized treatment approaches, enhancing the clinical management of EC. Compared to other published material, this review integrates recent molecular classifications and FIGO staging updates, presenting a more nuanced perspective on EC treatment.
As a literature review, the methodology involves synthesizing existing research. However, the authors should ensure the inclusion of meta analyses as much as possible in order to quantitatively synthesize the data from multiple studies, providing more robust conclusions.
The manuscript effectively addresses the main question by highlighting the potential benefits and challenges of integrating molecular pathology into EC treatment. The discussion on the future directions and the need for standardized testing protocols is particularly relevant.
The references are appropriate and cover a wide range of relevant studies. However, the authors should ensure that the most recent and pertinent studies are included to provide a current perspective on the topic.
However, I currently see no changes neccesarry and recommend it for acceptance in the current form.
Reviewer 2 Report
Comments and Suggestions for Authors
This paper is a systematic review of tailoring endometrial cancer treatment based on molecular pathology.
The paper is well written and the English language is appropriate and understandable.
The clinical topics are fascinating. The management of endometrial cancer is rapidly evolving because of the impact of new molecular assessment and classification on endometrial cancers.
The Authors carefully consider the possible local and systemic treatments including the current and future role of surgery, radiotherapy, and systematic therapy (chemotherapy and immunotherapy).
The Authors report thoroughly the results of the published studies focusing on the most significant prospective randomized clinical trials and retrospective, observational studies.
The possible future impact on fertility-sparing treatment is also taken into account.
Finally, the Authors are correctly worried by the significant disparities in endometrial cancer management worldwide.
